# A Study of the Feasibility of International ETS Cooperation between Shanghai and Korea from Environmental Efficiency and CO₂ Marginal Abatement Cost Perspectives

**Chao Qi *** and **Yongrok Choi ***

Global E-governance Program, Inha University, Inharo100, Nam-gu, Incheon 402-751, Korea

* Correspondence: dennischee@hotmail.com (C.Q.); yrchoi@inha.ac.kr (Y.C.);
  Tel.: +82-32-860-7760 (Y.C.); Fax: +82-32-876-9328 (Y.C.)

**Abstract:** With the worldwide spread of emissions trading schemes (ETSs) and the need for international cooperation on climate change, there is growing interest in linking ETSs. Along with sustainable development, preventing and controlling pollution, is now regarded as an urgent priority by China and Korea. In the context of the willingness of the Chinese and Korean governments to cooperate on ETS, this paper examines the feasibility of a pilot ETS cooperation between Shanghai and Korea from environmental efficiency and $CO_2$ marginal abatement cost (MAC) perspectives. We apply a directional distance function (DDF) and stochastic frontier analysis (SFA) to estimate the environmental efficiency and the $CO_2$ MAC of coal-fueled power plants in Shanghai and Korea using cross-sectional data from 2015. The results indicate that the group frontier environmental efficiency of Shanghai and Korea reached a similarly high score. However, as to meta-frontier environmental efficiency, the coal-fueled power plants in Korea performed better than those in Shanghai. The $CO_2$ MAC results indicate that, despite the small gap in efficiency performance, the $CO_2$ MAC of coal-fueled power plants is much higher than that in Shanghai due to the big feed-in tariff difference. This is because the MAC not only relates to the environmental efficiency, but also to the feed-in tariff. A higher feed-in tariff leads to higher MAC. To tackle this serious problem, which has also been addressed in previous studies, we suggest that policymakers should focus on the huge $CO_2$ MAC differences caused by feed-in tariff differences to avoid equity problems when building the structure of the Shanghai-Korea ETS cooperation. For instance, compared with power plants in Shanghai, policymakers should set a looser cap and a higher offset for Korean plants. To reduce the impact of feed-in tariff on carbon trading in the market, it would also be effective to arrange a higher quota or a lower carbon tax for coal-fueled power plants in Korea. In addition, policymakers should fill the gaps of 85.15% and 67.6% between the realistic market price and the MAC results of coal-fueled power plants in Shanghai and Korea, respectively, by introducing stricter regulations.

**Keywords:** directional distance function; stochastic frontier analysis; environmental efficiency; marginal abatement cost; international ETS cooperation

---

## 1. Introduction

From 30 November to 12 December 2015, more than 180 countries of the United Nations Framework Convention on Climate Change (UNFCCC) met in Paris to deal with greenhouse gas-emissions mitigation, adaptation, and finance in the 21st Conference of the Parties (COP21). As of March 2019, 195 members of the UNFCCC have signed the agreement, and 185 countries have become parties. The long-term goal of the Paris Agreement is to keep the rise in average global temperatures well

below pre-industrial levels of 2 °C and to limit warming to 1.5 °C, as this will greatly reduce the risk and impacts of climate change.

As to whether the Paris Agreement will bring about positive results, relevant studies are still inconclusive, especially concerning the specific implementation of the agreement in various countries. On the one hand, some studies suggest that the signing of the agreement alone represents a political success [1–3]. On the other hand, some studies have argued that the agreement is incomplete in some respects, due to the lack of enforcement mechanisms and provisions to compensate for losses caused by extreme weather events [4,5].

Since the global atmosphere is a public interest, regardless of national boundaries, addressing climate change requires effective international cooperation that needs to be integrated into a combination of strategic policy tools that include command-and-control (regulatory) and market-based policies. Carbon pricing and trade has become the preferred policy tool for reducing greenhouse gases (GHG) in many developed countries, and developing countries have also considered this kind of mechanism.

The European Union's emissions trading system, the first in the world, came into operation in 2005. After just 10 years, by 2015, 17 carbon trading systems have emerged around the world, covering 40% of the world's total GDP.

With the worldwide spread of ETS and the need for international cooperation on climate change, there is growing interest in linking ETSs. From this perspective, although the Paris Agreement did not set the rules of international carbon trading, it laid a foundation for the establishment of a carbon trading market mechanism, especially through international cooperation. The linkage of international ETSs can not only reduce the total emissions reduction cost of a linkage system and increase market liquidity and carbon price stability, but also promotes the integration of a separate carbon market, which is a bottom-up way to establish a global carbon market [6].

In 2014, a joint auction between California and Quebec became the world's first direct bilateral connection of ETSs, providing a good example for other regions to cooperate on carbon markets.

Under this climate strategy, there is an interaction of regional carbon regulations to reduce or eliminate the marginal cost differences between participants. The EU and Switzerland have signed an agreement to link their systems. Once the agreement has entered into force, linking would result in the mutual recognition of EU and Swiss emissions allowances. The EU and Australia have also considered the possibility of linking their systems. However, due to the repeal of the Australian system in 2014, linking negotiations have not been pursued [7]. In this way, a wider range of mitigation options can be obtained, thus improving overall efficiency [8,9]. International cooperation is expected to create a larger international market, which could increase market liquidity [10] and price stability [11].

There are different ways of cooperation through ETS at the international level. In a bilateral ETS connection, a carbon quota can be tradable between countries at a perfectly balanced price. If a country cannot reduce emissions at a lower cost, it can invest in new technologies for mitigation or buy an additional quota of emissions from another country that possesses a lower cost of emissions. Financial flows would go from the former country to the latter one that invests in emissions abatement technology [6].

Some factors may lead to technical barriers to the realization of international cooperation, including cap stringency, price management measure [12], and the recognition of offsets [8,13]. Cap stringency requirements, including cap setting, depend on the environmental and economic profile, or the level of development. In this sense, well-functioning cooperation also depends on how the strategy benefits the participants, without losing sight of the particularities, which include system compatibility, the nature of the system, and the existence of an absolute cap on emissions [14].

In order to solve economic and equity problems, comparable plans, climate policies, and long-term emissions trends should be considered, because these factors reflect the environmental plans and aggregate goals of the cooperation. Inadequate planning and unequal policies can lead to environmentally inefficient policies [8,15–18].

In principle, technical barriers do not include design scope, coverage, and other differences such as regulatory points and opt-in and opt-out provisions. In fact, two separate ETS markets cannot be identical in terms of different conditions [10,19]. Therefore, caution is necessary when linking two ETS markets from different sectors because it may lead to competition.

In line with this, the choice of the right partner is also an important consideration in the coordination of climate issues. In most cases, the chosen partner depends on geographical proximity, legal compatibility, potential distributional effects, respective ETS elements, and other factors [20]. Both EU&Switzerland ETS and California&Quebec ETS serve as good benchmarks for other regions that plan to collaborate in ETS.

In September 2016, China's national development and reform commission, Korea's Ministry of Strategy and Finance, and Japan's Ministry of the Environment held a meeting in Beijing and decided to hold a regular meeting on carbon emissions trading every year, through which they will discuss the establishment of a long-term unified market for carbon emissions trading and unified emissions testing methods, and seek a scheme for the exchange of carbon emissions among the three countries. As a more specific and substantive step, Korean ETS has signed a memorandum of understanding with China's Beijing Environment Exchange. Under this agreement, China and Korea will exchange market information and share experience on ETS operations. They agreed to expand cooperation on ETS, including joint efforts to link ETS in these two countries. ETS cooperation between these two countries can be expected in the future.

The intense interest in ETS cooperation between China and Korea, in addition to the geographical proximity, is mainly due to their large international trade volume. Korea's carbon market is the first ETS among developing countries, while China is the world's largest carbon trading market. Both the China ETS and the Korea ETS received technological and institutional support from the EU ETS, which provided the favorable conditions for compatibility between these two ETS markets [7].

China is the second-largest economy and the country with the largest population in the world. It is also the largest emitter of carbon dioxide, accounting for 27% of the world's emissions. In order to fulfill its commitment to the Paris Agreement and improve the environment, China has set a target of reducing its carbon dioxide emissions by 60–65% (from the 2005 levels) by 2030. As a stepwise measure, China set up seven pilot carbon markets in 2011, including five cities of Shanghai, Beijing, Tianjin, Chongqing, and Shenzhen, and two provinces of Guangdong and Hubei. Furthermore, in December 2017, China released a plan for a nationwide carbon emissions market, using the power generation industry as a pilot industry. Note that power generation, especially coal-fired power plants, accounts for 47% of China's total coal consumption and 34% of its carbon dioxide emissions [21]. China has shown willingness to explore cooperation and docking of regional international ETS systems after the successful establishment of the nationwide ETS system. China has great potential to play a leading role in the international ETS market [6]. In July 2018, China and EU released the China-EU leaders' statement on climate change and clean energy that will exchange experience and promote cooperation on ETS. In June 2017, Chinese government and the governor of California discussed the possibility of linking their ETS. Both sides promised that they would study and overcome the obstacles and complexities of the cooperation constantly. Similarly, China and Korea have exchanged in-depth views on ETS cooperation issues.

A neighbor of China, Korea is the 11th largest economy in the world, accounting for about 2.2% of the world's total primary energy consumption in 2017, making it the eighth largest energy consumer in the world. In addition, Korea accounted for 2.3% of global coal consumption, ranking sixth in the world in 2017 [22]. In 2017, the $CO_2$ emissions of Korea were 679.7 million tons, ranking fourth among the Organization for Economic Co-operation and Development (OECD) nations after the United States (5.0877 billion tons), Japan (1.1766 billion tons), and Germany (763.8 million tons), and representing an increase of y 24.6% from the 2007 level. The speed of $CO_2$ emissions increase of Korea ranked second among the OECD countries after Turkey's (50.5%). According to the analysis, South Korea's high dependence on coal is the main reason for its high carbon dioxide emissions, as well as the

main cause of air pollution and haze [23]. In 2009, the Korean government enacted a "low-carbon green growth" national carbon emissions reduction policy. In 2015, Korean government passed a law requiring to reduce Korea's carbon dioxide emissions by 37% below business as usual (BAU) levels by 2030. Moreover, in the same year as the Paris Agreement was signed, Korea formally launched a national ETS as a concrete measure to reduce $CO_2$ emissions.

Due to the huge differences in economic scale, population size, economic development stage, and social and legal systems between China and Korea, it is unrealistic to pursue international ETS cooperation at a national level in the short term. Following China's approach of establishing pilot regions and pilot industries in the development of a carbon market, China and Korea should conduct pilot international cooperation in areas of similar economic scale and development stage.

Shanghai is one of the pilot regions for a carbon trading market and the largest city in China. In addition, the national carbon market trading center of China was established in Shanghai because of its developed financial industry base, as well as its special status in China's political and economic arena, and its good performance as a pilot carbon market. In terms of carbon market maturity, economic development level, and market size, Shanghai is closest to Korea, and thus the most appropriate region to cooperate with Korea on ETS. Moreover, given that the charcoal power industry plays the most important role in the carbon market in both China and Korea, the coal industry is the first choice for the regional cooperation pilot project.

As of 2015, there are systems operating in jurisdictions that vary largely in terms of their geographical scope, economic profile, and energy mix. Indeed, when it comes to ETS, there is no one-size-fits-all approach; rather, the variability and individuality of the ETS are the key to its success. As a pilot international ETS cooperation, the Shanghai-Korea ETS market should explore an adaptable mechanism.

Table 1 describes the transaction situation of the Shanghai and Korea ETSs in 2015 [24,25]. In terms of transaction sums, the Korea ETS is 19.32% higher than the Shanghai ETS, but in terms of transaction volume, the Shanghai ETS is about three times higher than the Korean ETS. Accordingly, in terms of average market price, the Korean ETS is about four times higher than the Shanghai ETS. Regardless of which ETS is being discussed, the power industry plays an important role in the dominant position, accounting for 50.2% and 48.8%, respectively, of total transaction volume. Note that in both Shanghai and Korea, although there are no separate transaction data on coal-fired power plants, they occupy an important position in the entire power generation industry, accounting for 67.9% and 42%, respectively, of the total power generation in 2016 [26].

**Table 1.** The transaction situations of the Shanghai and Korea ETSs in 2015.

|  | Shanghai | Korea |
|---|---|---|
| Sums (millions of U.S. dollars) | 46.18 | 55.01 |
| Volume (KT) |  | 5734 |
| Average market price (U.S. dollars/ton) | 2.53 | 9.59 |
| Volume of power industry | 9156 | 2800 |
| Volume ratio of power industry | 50.2% | 48.8% |

Source: Shanghai ETS annual report 2015 (http://www.cneeex.com/). Korea ETS annual report 2015 (http://www.energy.or.kr/web/kem_home_new/new_main.asp).

Over the past two decades, many empirical studies have examined the efficiency and pollutant MAC of coal-fueled power plants in China and Korea, on account of the important global position of these two countries and the key role of coal-fueled power plants in the power industry [27–32].

In this paper, we make a contribution to the literature by extending the approach of previous studies. We introduce DDF and SFA, which not only seek to maximize desirable outputs while reducing undesirable outputs, but also can differentiate the inefficiencies and random errors. Moreover, we introduce Meta-frontier to solve group heterogeneity problems. Through environmental efficiency

and pollutant MAC perspectives, we first attempt to make an analysis of the feasibility of the international ETS cooperation between China and Korea.

The rest of this paper is organized as follows: Section 2 summarizes the trend of the method to estimate efficiency and pollutants' MAC and related studies on international ETS cooperation. Section 3 then describes the methodology. Section 4 uses the proposed approach to empirically analyze the environmental efficiency and pollutants' MAC results and the feasibility of international ETS cooperation between China and Korea, and Section 5 concludes with some policy implications.

## 2. Literature Review

In the early stages of pollutants' MAC studies, Aigner et al. and Schmidt estimated pollutants' MAC by the deterministic production function [33,34]. Since then, Pollak et al. and Gollop et al. have begun to estimate pollutants' MAC by the deterministic cost function [35,36]. After Pittman creatively estimated pollutants' MAC by Shephard distance function, this kind of method based on distance function (DF) begun to be used broadly [37,38]. Compared with DF, the directional distance function (DDF) is a relatively new methodology for pollutants' MAC or environmental efficiency estimation that seeks to maximize desirable outputs while simultaneously reducing undesirable outputs. For instance, Fare et al. estimated U.S. coal-fueled power plants' efficiency performance. Zhang et al. estimated the $CO_2$ emissions performance of Korean coal-fueled power plants. Both of them used DDF [30].

The approaches to solving distance functions include non-parametric methods and parametric methods. DEA is the most widely used non-parameter method and is a quantitative analysis method based on multiple inputs and output indexes, in which linear programming method is used to evaluate the relative efficiency among comparable units. The advantage of the DEA method is that it does not need to set a specific functional form for the underlying technology in advance [39]. By the DEA approach, Wang et al. measured the $CO_2$ MACs for 28 Chinese provinces. Liu et al. measured the $CO_2$ MACs for 30 Chinese provinces from 2005 to 2007, and Zhang et al. measured the $CO_2$ MACs for 29 Chinese provinces from 2006 to 2010 [39–41]. However, the DEA method can only be applied when the function is differentiable everywhere. Moreover, there are different slopes of the efficient observation located on the frontier, and a different choice of slope will lead to different results [42].

The linear programming (LP) method is one of the parametric approaches to solve distance functions. By the LP approach, Du et al. measured the $CO_2$ MACs for 29 Chinese provinces from 1995 to 2007 [43]. However, the LP method ignores random error and statistical noise [39].

In order to avoid the weaknesses of the DEA and LP approaches, we introduce the stochastic frontier approach (SFA). The SFA not only differentiates between inefficient parts and random error, but also considers statistical noise. For instance, Wei et al. estimated the $CO_2$ MACs among China's coal-fueled power plants by the SFA approach [44]. Choi and Qi estimated the MAC of 50 coal-fueled power plants in Korea by the SFA approach [45]. Table 2 summarizes some representative studies on MAC estimation in recent years.

**Table 2.** Studies on MAC estimation.

| Studies | Year | Method | Objective |
|---|---|---|---|
| Wang et al. | 2011 | DDF/DEA | $CO_2$ MAC of 28 provinces in China |
| Liu et al. | 2011 | DF/DEA | $CO_2$ MAC of 30 provinces in China |
| Du et al. | 2012 | DF/LP | $CO_2$ MAC of 29 provinces in China |
| Zhang et al. | 2014 | DDF/LP | $CO_2$ MAC of 29 provinces in China |
| Wei et al. | 2013 | DDF/SFA | $CO_2$ MAC of the coal-fueled power plants in China |
| Oh et al. | 1999 | DF/LP | Airborne pollutants MAC of Korea's power plants |
| Lee | 2010 | DF/LP | $CO_2$ MAC of Korea's coal-fueled power plants |
| Choi and Qi | 2019 | DDF/SFA | $CO_2$ MAC of 50 coal-fueled power plants in Korea |

Previous studies estimating the pollutants' MAC or efficiency performance of power plants have focused on specific groups. However, if the group heterogeneity of coal-fuel power plants is not taken

into account, the estimated efficiency score or MAC results may be biased as the heterogeneity may lead to differences in production techniques.

O'Donnell et al. indicated that efficiency under meta-frontier technology can be decomposed into group efficiency and meta-technology ratio (MTR). Group efficiency measures the relative efficiency of observation under specific group-frontier technology, while MTR (also known as the technology gap ratio) measures the distance between the group-frontier technology and the meta-frontier technology [46]. O'Donnell et al. pointed out that meta-frontier technology does not exceed group technology, so the meta-technology ratio is not greater than 1. The higher the meta-technology ratio, the closer the group frontier technology is to the meta-frontier technology. If the meta-frontier technology value equals 1, there is no gap between the two technologies. That is, the two technologies overlap completely [47]. Following O'Donnell, we divided the coal-fueled power plants of Shanghai and Korea into two groups to solve the group heterogeneity problem.

The issue of international ETS cooperation has attracted an enormous amount of research, especially after the Paris Agreement. The EU ETS, the first international emissions trading system in the world, was set up in 2005. It is also the biggest one, accounting for over three-quarters of international carbon trading [48]. Therefore, most previous empirical studies focused on EU ETS. For instance, Chapman, Zetterberg, and Marschinski estimated the impact of the ETS cooperation between EU and USA [13,49]. They found that system stringency is a serious sticking point, as market realities will prevent offsets from stymieing linking negotiations. However, the opportunity should be seized by the USA to engage in massive efforts to take part in the EU's firm stance on emissions reductions through tighter caps, less offset use, and a lower price ceiling. Marschinski et al. and Hubler et al. investigated a proposal for the cooperation between the EU ETS and the China ETS [50,51]. They revealed that linking the Chinese ETS to the European ETS and restricting the transfer volume to one-third of the EU's reduction effort creates at best a small benefit for China, yet with smaller sectoral output reductions than auctioning. These results are evidence of the importance of designing the Chinese ETS wisely. By simulating autarky and linkage scenarios, Gavard et al. estimated a sectoral ETS on energy-intensive industries in the EU, the USA, and China [52]. They found that the limit results in different carbon prices between China and Europe or the USA. Although the impact on low-carbon technologies in China is moderate, global emissions reductions are more significant than in the absence of international trading due to reduced carbon leakage. If China captures the rents associated with limited permit trading, they show that it is possible to find a limit threshold that makes both regions better relative to carbon markets operating in isolation. In addition, Xu et al. implemented a different scenario analysis and simulated the establishing of a conceivable multi-region integrated emissions trading scheme with China, the USA, Europe, Australia, Japan, and South Korea by utilizing a computable general equilibrium model [53]. They found that the integration of ETS results in the redistribution of clean energy in the participating countries. In addition, it is worth noting that the multi-region integrated ETS would have significant impacts on the role each region plays in international trade, leading to an 11% decline in net exports for China in the MR scenario compared with the SR scenario. After summarizing the above previous studies, we find that, by optimizing the allocation of emissions permits, the integration of ETS will have complex economic and energy implications for different participants. For some countries, economic benefits are expected, while their expansion into the clean energy industry will slow down. On the other hand, such integration may also promote the development of clean energy in other countries, while adversely affecting their international competitiveness. It is worth noting that, before participants join the multi-regional comprehensive ETS mechanism, in addition to addressing political, institutional, and technical barriers, they need to clarify their roles and balance the interests of all participants.

To date, considering that they are important countries in terms of their economy and $CO_2$ emissions, the international ETS cooperation between China and Korea has not attracted adequate attention in empirical studies. Therefore, we will fill this gap from environmental efficiency and pollutants' MAC

perspectives by the DDF and SFA methods; meanwhile, we will introduce a meta-frontier analysis to differentiate the group heterogeneity.

## 3. Methodology

According to Chung et al., the pollution problem in SFA can be expressed such that desirable outputs are often accompanied by undesirable outputs [54]. In our case, when a coal-fueled power plant produces electricity as a desirable output, $CO_2$ will also be produced as an undesirable output. We assume that inputs are denoted as $x \in R_+^N$, with desirable outputs as $y \in R_+^N$ and undesirable outputs as $b \in R_+^N$, so the meta-frontier can be defined by the output sets as follows:

$$S = \{(x, y, b) : x \text{ can produce } (y, b)\}. \tag{1}$$

The meta-frontier S is defined as a boundary that envelopes all the observations, which describes how the input vector $x$ can produce a set of desirable output and undesirable output ($y$, $b$).

Following Fare et al., we assume that the desirable output $y$ and undesirable output $b$ are produced jointly [55]. This assumption also satisfies the following three properties:

(i) The desirable outputs and the undesirable outputs are null-joint. If ($y$, $b$) $\in$ P($x$) and $b = 0$ then $y = 0$, which indicates that the desirable and undesirable outputs are produced simultaneously; in other words, if there are no undesirable outputs, there will be no desirable outputs.

(ii) If ($y$, $b$) $\in$ P($x$) and $0 \leq \theta \leq 1$ then ($\theta y$, $\theta b$) $\in$ P($x$), which indicates that the desirable and undesirable outputs are jointly produced under the weak disposability. In other words, if the undesirable outputs decrease, the desirable outputs will decrease proportionately and simultaneously. The reduction of undesirable outputs is costly.

(iii) If $x' \geq x$ then P($x'$) $\supseteq$ P($x$), which implies that the inputs and the outputs increase simultaneously, that is to say, the inputs represent strong disposability.

Then we can define the meta directional output distance function as

$$\vec{D}(x, y, b; g_y, -g_b) = \max\{\lambda : (y + \lambda g_y, b - \lambda g_b) \in S\}, \tag{2}$$

where $g = (g_y, g_b)$ represents the directional vector. By this function, the desirable outputs can be expanded maximally on the direction $g_y$ while contracting the undesirable outputs on the direction $g_b$. Additionally, here $\lambda$ implies the technical cap between observations and the meta-frontier.

In our study, constrained by resources and regulatory or other environmental factors, coal-fueled power plants in China and Korea are divided into two groups due to the existence of sub-technology sets, where the group-frontier can be defined as

$$S^k = \{(x, y, b) : x \text{ can produce}(y, b)\}, k = 1, 2. \tag{3}$$

Thus the $k$th group directional output distance function can be represented as:

$$\vec{D}^k(x, y, b; g_y, -g_b) = \max\{\lambda : (y + \lambda g_y, b - \lambda g_b) \in S^k\}. \tag{4}$$

Furthermore, according to Zhang et al. and Du et al., we assume all group production sets belong to $S$, and define $S = (S_1 \cup S_2)$, which is demonstrated in Figure 1 [30,32]. We assume that AB is the group environmental efficiency (GEE) measured by group frontier, and AC is the meta-frontier environmental efficiency (MEE), measured by the meta-frontier; BC is the meta-technology ratio (MTR) between the group frontier and the meta-frontier.

According to Bai et al. (2016), the aforementioned meta and group directional output distance function have five properties [56]:

(i) $\vec{D}(x, y, b, g) \geq 0 \; \forall \; (y, b) \in S(x)$, if the directional output distance is greater than 0, the observation is inefficient; if the observation is located on the frontier $S(x)$, the directional distance value is 0.

(ii) $\vec{D}(x, y', b; g) \geq \vec{D}(x, y, b; g) \; for \; (y', b) \geq (y, b) \in S(x)$, if a power plant produces more desirable outputs and the same undesirable outputs with the same inputs, the inefficiency will not increase.

(iii) $\vec{D}(x, y, b'; g) \geq \vec{D}(x, y, b; g) \; for \; (y, b') \geq (y, b) \in S(x)$, similarly if a power plant produces more undesirable outputs and the same desirable outputs with the same inputs, the inefficiency will not increase.

(iv) $\vec{D}(x, \theta y, \theta b; g) \geq 0 \; for \; (y, b) \in S(x) \; and \; 0 \leq \theta \leq 1$. implies that the desirable outputs and the undesirable outputs are have joint weak disposability.

(v) $\vec{D}(x, y, b; g)$ is concave in $(y, b) \in S(x)$. This property determines the sign of the output elasticity of substitution.

Additionally, the meta-directional distance function satisfies the translation property below, which is convenient for calculation:

$$D_0(x, y + \alpha g_y, b - \alpha g_b; g) = D_0(x, y, b; g) - \alpha. \tag{5}$$

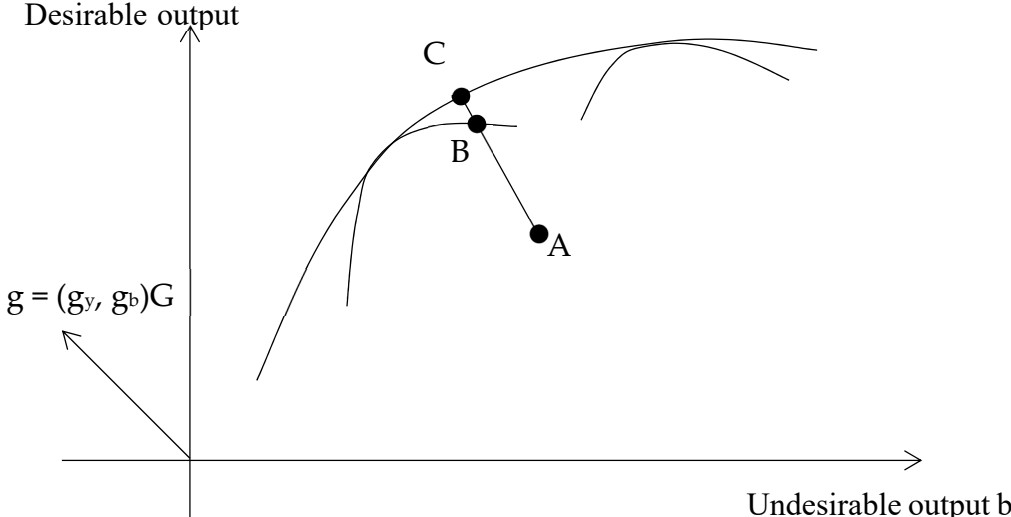

**Figure 1.** Relationship between meta-frontier and group frontier.

On the production frontier, the inefficiency value is 0, so we set 0 as the dependent variable. We get the equation below if we put DDF into the SFA model:

$$0 = D_0(x, y, b, g) + v - u. \tag{6}$$

As for the meta-frontier, the concavity property should be satisfied. The concavity implies that, under fixed inputs, if the desirable outputs increase by $\alpha g_y$ and the undesirable outputs decrease by $\alpha g_b$, then the value of the function will be reduced by $\alpha$.

As depicted in Figure 1, there is a meta-technology ratio (MTR) between the meta-frontier inefficiency (AC) and group frontier inefficiency (AB) because the efficiency estimated by the meta directional output distance function is based on all groups.

$$MTR_k = (1 - \vec{D}(x_k, y_k, b_k; g)) / (1 - D(x_k, y_k, b_k; g)) \tag{7}$$

The MTR measures the ratio of the environmental efficiency of a certain group stochastic function relative to the potential environmental efficiency that is defined by the meta-frontier function. The MTR

implies the technology difference for the given group according to the currently available technology for power plants in that group, and can be expressed as follows:

$$MTR_k = \frac{GEE^k}{MEE}. \tag{8}$$

To solve the SFA problems, we need a specific functional form. In our study, we estimate the directional output distance function by the quadratic functional, which satisfies the translation property. By expanding Equation (6), we get the equation below:

$$
\begin{aligned}
D_0(k_{it}, l_{it}, e_{it}, b_{it}; g) = \quad & \beta_0 + \beta_k k_{it} + \beta_e e_{it} + \beta_y y_{it} + \beta_c b_{it} \\
& + \beta_{kl} k_{it} l_{it} + \beta_{ke} k_{it} e_{it} + \beta_{ky} k_{it} y_{it} + \beta_{kc} k_{it} b_{it} \\
& + \beta_{le} l_{it} e_{it} + \beta_{ly} l_{it} y_{it} + \beta_{lc} l_{it} b_{it} + \beta_{ey} e_{it} y_{it} \\
& + \beta_{ec} e_{it} b_{it} + \beta_{yc} y_{it} b_{it} + \tfrac{1}{2}\beta_{kk}(k_{it})^2 + \tfrac{1}{2}\beta_{ll}(l_{it})^2 \\
& + \tfrac{1}{2}\beta_{ee}(e_{it})^2 + \tfrac{1}{2}\beta_{yy}(y_{it})^2 + \tfrac{1}{2}\beta_{cc}(b_{it})^2 + v_i + u_i
\end{aligned} \tag{9}
$$

By taking the undesirable output b as the dependent variable, we get the following equation:

$$
\begin{aligned}
-b_i = \quad & \beta_0 + \beta_k k_i + \beta_l l_i + \beta_e e_i + \beta_y(y_i + b_i) + \beta_{kl} k_i l_i \\
& + \beta_{ke} k_i e_i + \beta_{ky} k_i(y_i + b_i) + \beta_{le} l_i e_i + \beta_{ly} l_i(y_i + b_i) \\
& + \beta_{ey} e_i(y_i + b_i) + \tfrac{1}{2}\beta_{kk}(k_i)^2 + \tfrac{1}{2}\beta_{ll}(l_i)^2 + \tfrac{1}{2}\beta_{ee}(e_i)^2 \\
& + \tfrac{1}{2}\beta_{yy}(y_i + b_i)^2 + v_i - u_i
\end{aligned} \tag{10}
$$

$$\beta_y - \beta_b = -1, \beta_{yy} = \beta_{bb} = \beta_{yb}, \beta_{kb} = \beta_{ky}, \beta_{lb} = \beta_{ly}, \beta_{ec} = \beta_{ey},$$

where $y$ and $b$ represent the desirable output (electricity) and the undesirable output ($CO_2$ emissions), respectively. In addition, $k$, $l$, and $e$ represent capital input, labor input, and energy input, respectively. $g$ represents the direction vector, $i = 1, 2, ..., N$ represents each power plant. $v_i$ represents the random errors and $u_i$ represents the inefficiency.

Following Fare et al. and Chung, we estimate the MACs of each power plants by the equation that follows [54,55]:

$$q_i = -p_i \frac{\partial D_i(k_i, l_i, e_i, y_i, b_i; 1, -1)/\partial b_i}{\partial D_i(k_i, l_i, e_i, y_i, b_i; 1, -1)/\partial y_i}, \tag{11}$$

where $q_i$ represents the value of MACs and $p_i$ represents the feed-in tariff.

We get Equations (12) and (13) if we put the specific form of the directional distance function into Equation (11):

$$\frac{\partial D_i(k_i, l_i, e_i, y_i, b_i; 1, -1)}{\partial b_i} = \beta_b + \beta_{kb} k_i + \beta_{ib} l_i + \beta_{eb} e_i + \beta_{yb} y_i + \beta_{bb} c_i \tag{12}$$

$$\frac{\partial D_i(k_i, l_i, e_i, y_i, b_i; 1, -1)}{\partial y_i} = \beta_y + \beta_{ky} k_i + \beta_{ly} l_i + \beta_{ey} e_i + \beta_{yb} c_i + \beta_{yy} y_i. \tag{13}$$

Thus, the equation for estimating MAC can be denoted as:

$$q_i = -p_i \frac{\beta_b + \beta_{kb} k_i + \beta_{lb} l_i + \beta_{eb} e_i + \beta_{yb} y_i + \beta_{bb} c_i}{\beta_y + \beta_{ky} k_i + \beta_{ly} l_i + \beta_{ey} e_i + \beta_{yb} c_i + \beta_{yy} y_i}. \tag{14}$$

By calculating the coefficient value of each variable in the above equations, the environmental efficiency and emissions reduction cost of each power plant can be estimated.

## 4. Empirical Results

### 4.1. Data and Variables

As shown in Table 1, we selected 92 coal-fueled power plants in Shanghai and 53 coal-fueled power plants in Korea as our study sites. These cover all the coal-fueled power plants with an installed capacity above 10 MW in Shanghai and Korea. The data came from the Shanghai electric power corpus (2015) and the Korea Electric Power Corporation's Statistics of Electric Power (2015) [21,57]. We used the installed capacity as capital input (k), the coal consumption as the energy input (e), the number of employed staff as the labor input (l); these were our three input variables. Additionally, we used each power plant's gross electricity generation as the desirable output (y) and carbon dioxide emissions as the undesirable output (b); these were our two output variables. The data on electricity prices (P) are collected from the feed-in tariff of each coal-fired power plant of Shanghai and Korea in 2015.

However, there are no carbon dioxide emissions data available. Therefore, by following the criteria published on the website of the International Panel on Climate Change and the NRDC's National Coordination Committee Office on Climate Change and Energy Research Institute, we calculated $CO_2$ as follows [58]:

$$CO_2 = E \times CF \times CC \times COF \times (44/12) \tag{15}$$

where *E* indicates the amount of consumption of coal, *CF* indicates the transformation factor (tons of carbon/TCE), CC indicates the carbon content, and *COF* indicates the carbon oxidation factor, which estimates the percentage of actual carbon when combustion happens. The figure "44/12" indicates the ratio of the weight of one carbon atom combined with two oxygen atoms to the weight of a carbon atom. The '*CF*' reported by the Korean Ministry of Knowledge Economy is used to estimate the $CO_2$ emissions.

All data are based on the year 2015 and thus the empirical experiment is a cross-sectional comparative analysis between Korea and Shanghai. As shown in Table 3, the average scale of coal-fueled power plants in Korea is larger than that in Shanghai, while the standard deviation is similar.

**Table 3.** Descriptive statistics.

| Variable | Obs | | | Mean | | | Std. Dev. | | |
|---|---|---|---|---|---|---|---|---|---|
| Region | SH | KO | Tot | SH | KO | Tot | SH | KO | Tot |
| Capital (MW) | 92 | 53 | 145 | 182.33 | 399.47 | 261.69 | 221.81 | 150.97 | 224.35 |
| Energy (KT) | 92 | 53 | 145 | 280.02 | 2811.09 | 1205.16 | 380.25 | 971.56 | 1388.62 |
| Labor (Per Person) | 92 | 53 | 145 | 104.27 | 297.55 | 174.91 | 104.16 | 101.67 | 138.97 |
| Electricity (kMWh) | 92 | 53 | 145 | 791.97 | 3889.35 | 1924.12 | 1261.28 | 1403.68 | 1989.27 |
| $CO_2$ (KT) | 92 | 53 | 145 | 672.06 | 2815.58 | 1455.55 | 912.59 | 966.67 | 1391.65 |

### 4.1.1. Empirical Results and Discussion

Based on Equation (10), we solved the SFA problem by incorporating all the parameter estimate coefficients, as shown in Table 4, into the Stata software program (College Station, TX, USA). We used the maximum likelihood method to estimate the stochastic frontier. All the variables obtained are consistent with expectations and passed the statistical test. If the sign is negative, the negative effect of the variable in the production process is greater than the positive effect; on the contrary, if the sign is positive, the positive effect of the variable in the production process is greater than the negative effect.

**Table 4.** Parameter estimates.

| Parameter | Coef. (Std. Err.) | | |
|---|---|---|---|
| | **Group Frontier (Shanghai)** | **Group Frontier (Korea)** | **Meta-Frontier** |
| $\beta_l$ | $4.23 \times 10^{-6}$ ($1.22 \times 10^{-6}$) *** | $-0.0009125$ ($0.0003922$) *** | $0.0004164$ ($0.0000958$) *** |
| $\beta_k$ | $-2.41 \times 10^{-6}$ ($2.11 \times 10^{-6}$) | $0.0041788$ ($0.0011684$) *** | $0.0003978$ ($0.0001042$) *** |
| $\beta_e$ | $-1.000032$ ($0.0000229$) *** | $-0.3308924$ ($0.1548403$) ** | $0.1499747$ ($0.0252257$) *** |
| $\beta_y$ | $0.000014$ ($0.0000107$) | $-0.4943289$ ($0.1222343$) *** | $-0.5134027$ ($0.0146761$) *** |
| $\beta_{ll}$ | $9.91 \times 10^{-7}$ ($2.90 \times 10^{-7}$) *** | $4.38 \times 10^{-6}$ ($6.47 \times 10^{-6}$) | $-9.88 \times 10^{-7}$ ($3.71 \times 10^{-7}$) *** |
| $\beta_{kl}$ | $0.0000124$ ($2.50 \times 10^{-6}$) *** | $-0.000015$ ($6.69 \times 10^{-6}$) *** | $-6.47 \times 10^{-7}$ ($3.50 \times 10^{-7}$) ** |
| $\beta_{le}$ | $-0.0001162$ ($0.0000233$) *** | $-0.0000787$ ($0.0019222$) *** | $0.0006612$ ($0.0001256$) *** |
| $\beta_0$ | $1.49 \times 10^{-6}$ ($1.03 \times 10^{-6}$) | $-0.0269538$ ($0.0137456$) ** | $-0.0549784$ ($0.0063817$) *** |
| $\beta_{kk}$ | $3.41 \times 10^{-6}$ ($8.35 \times 10^{-6}$) | $-8.71 \times 10^{-6}$ ($2.68 \times 10^{-6}$) * | $-5.29 \times 10^{-6}$ ($5.78 \times 10^{-7}$) *** |
| $\beta_{ke}$ | $-0.0000471$ ($0.0000204$) *** | $0.0087048$ ($0.006622$) ** | $-0.000929$ ($0.0002616$) *** |
| $\beta_{ee}$ | $0.0008673$ ($0.0002042$) *** | $2.425098$ ($1.49435$) | $-1.160946$ ($0.0957935$) *** |
| $\beta_{yy}$ | $0.0001582$ ($0.0000413$) *** | $0.8976776$ ($0.53042$) ** | $-0.1145199$ ($0.0274249$) *** |
| $\beta_{ly}$ | $0.00005$ ($0.0000101$) *** | $0.0009055$ ($0.001164$) | $-0.0002271$ ($0.0000555$) *** |
| $\beta_{ky}$ | $0.0000196$ ($7.86 \times 10^{-6}$) *** | $-0.0029704$ ($0.0024377$) | $0.0005836$ ($0.0001162$) *** |
| $\beta_{ey}$ | $-0.0003704$ ($0.0000918$) *** | $-1.644005$ ($0.9839597$) ** | $0.375444$ ($0.051916$) *** |

Note: The standard deviation is provided in parentheses; * $p < 0.1$; ** $p < 0.05$; *** $p < 0.01$.

Table 5 shows the results for environmental efficiency under group frontier technologies and meta-frontier technologies. As shown in this table, the average GEE value of coal-fueled power plants in Shanghai is 0.9289, indicating that, on average, coal-fueled power plants in Shanghai can increase their environmental efficiency by approximately 7.11% if all of them operate by group production technologies. Similarly, the average GEE value of coal-fueled power plants in Korea is 0.9218, which indicates that, on average, coal-fueled power plants in Korea can increase their environmental efficiency by approximately 7.82% if they operate by group production technologies. We found that the group environmental efficiency results of coal-fueled power plants in both Shanghai and Korea reached a high score. Their good performance in terms of environmental efficiency is a result of the sustainable development policies in these two countries and the exploration of new technology for improving production efficiency. From 2007, the shift of economic growth mode toward green growth or sustainable development became one of the most important national policies of China, and the power industry, as the key sector of pollutant emissions, started to heavily invest in key technologies for ultra-low emissions. In 2009, the Korean government set up the "green economic growth strategy" and in 2005 Korea established ETS, which was the first ETS among developing countries. Because coal-fueled power plants play an important role in total $CO_2$ emissions, this sector has been under considerable pressure to improve its environmental efficiency.

**Table 5.** Environmental efficiency results.

| Shanghai | | | Korea | | |
|---|---|---|---|---|---|
| **GEE** | **TGR** | **MEE** | **EE** | **TGR** | **MEE** |
| 0.9289 | 0.8450 | 0.7849 | 0.9218 | 0.8704 | 0.8023 |

The MEE value of the coal-fueled power plants in Korea is 0.8023, 2.22% higher than in Shanghai. The TGR value of coal-fueled power plants in Korea is 0.8704, 3% higher than in Shanghai. This indicates that the environmental efficiency of coal-fueled power plants in Korea is closer to the meta-frontier technology. Given the different stages of development in Korea and China, there is a gap of the environmental efficiency between the coal-fueled power plants in these two regions, but the gap is not

huge. Because the coal-fueled power plants in Shanghai have benefited from heavy investments in ultra-low emissions technology over the last decade.

Table 6 shows the $CO_2$ MAC results of coal-fueled power plants in Shanghai and Korea. The average MAC result is $17.04 and $29.6 in Shanghai and Korea, which is much higher than the real market price shown in Table 2. In other words, the market price only reflected 14.85% and 32.4%, respectively, of the $CO_2$ MAC results. In both ETSs, coal-fueled power plants frequently act as a buyer in the market because they can purchase quotas at a very low price; this means that they can emit $CO_2$ at a very low cost. So, as a matter of fact, the extremely low market price cannot stimulate the coal-fueled power plants to invest further in technology and improve environmental efficiency. The standard deviation of the MAC results of coal-fueled power plants in Korea is higher than that in Shanghai. These results indicate that, theoretically, coal-fueled power plants in Korea can reduce $CO_2$ emissions more efficiently. We also found that there is a considerable overlap between the MAC range of coal-fueled power plants in these two regions, which makes the power plants in both countries act as quota sellers or quota buyers, rather than all of the power plants in one region acting as quota sellers while all of the power plants in another region act as quota buyers. This fact will stimulate their efficiency performance and help ensure effective carbon emissions reductions for power plants in both regions, with no one-sided benefits. In addition, although the gap in environmental efficiency between coal-fired power plants in Shanghai and those in Korea is small, the MAC results' difference reaches 73.7%. Obviously, this significant difference in the MAC result is not due to the gap in environmental efficiency, but to the difference in feed-in tariffs between the two regions. The MAC not only relates to the environmental efficiency, but also to the feed-in tariff. A higher feed-in tariff leads to a higher MAC. If international ETS cooperation is to be achieved, differences in feed-in tariffs must be taken into account to avoid equity problems. For instance, policymakers should arrange a higher quota or lower carbon tax for coal-fueled power plants in Korea to reduce the impact of feed-in tariffs on carbon trading in the market.

**Table 6.** $CO_2$ MAC results (in U.S. dollars/ton).

| Coefficient | Shanghai | Korea | Total |
|:---:|:---:|:---:|:---:|
| Obs | 92 | 53 | 145 |
| Mean | 17.04 | 29.60 | 21.63 |
| Std. Dev. | 5.35 | 7.73 | 8.59 |
| Min | 7.32 | 18.96 | 7.32 |
| Max | 37.63 | 51.45 | 51.45 |

## 5. Conclusions

Along with the sustainable developmental strategy, the question of preventing and controlling pollution is regarded as urgent by both China and Korea. Indeed, how to effectively protect the environment has become one of the thorniest issues of this century.

The Paris Agreement, signed in 2015, laid the foundation for the establishment of carbon trading market mechanisms, especially through international cooperation. Compared with sectoral or domestic ETS, international coordination can improve the efficiency of ETS, while substantially reducing emissions, to compensate for sector-specific costs of ETS and ultimately improve political acceptability. Harmonization may have the potential to avoid distributional effects and contribute to the environmental and economic benefits.

China and Korea are two major $CO_2$ emitters, together accounting for about 30% of the total $CO_2$ in the world. The two countries are not only geographically close, but also have a huge amount of trade volume with each other. In addition, both governments have shown willingness to cooperate on ETSs. Therefore, it is necessary to investigate the feasibility of international ETS cooperation between them. To avoid differences in economic scale, population size, and economic development stage, we selected Shanghai as a pilot region to cooperate with Korea. First of all, Shanghai is the most

developed region in China and the closest to the stage of Korea's economic development. Second, the formation of Shanghai's metropolitan area is of special importance in China. It not only promotes economic development and the district integration in the Yangtze Delta, but also serves as a model for the development of the whole country. Third, the national carbon market trading center of China was established in Shanghai, because of its developed financial industry base, as well as its special status in China's political and economic arena, and its good performance as a pilot carbon market.

The environmental efficiency results from coal-fueled power plants in Shanghai and Korea revealed that the group environmental efficiency is at a high level in both Shanghai and Korea, reaching 0.9289 and 0.9218, respectively, indicating that green economic policies and the establishment of ETS have played a positive role in these two regions and the exploration and application of new technology have been very successful. However, as to the meta-frontier environmental efficiency results, the value of coal-fueled power plants in Korea is 0.8023, which is higher than that in Shanghai. This result indicates that coal-fueled power plants in Shanghai are more likely to improve their environmental efficiency through benchmarks in international ETS cooperation. Large-scale ultra supercritical pressure units have been put into operation in Korea; meanwhile, the government and companies in Korea have invested significantly in carbon capture and storage technology.

This study also estimated the $CO_2$ MAC of coal-fueled power plants in Shanghai and Korea. The results revealed that the real market price is much lower than our MAC results. In the Shanghai-Korea international ETS market, policymakers should impose a carbon tax, adjust their carbon emissions quotas, and introduce more regulatory measures—especially for coal-fueled power plants—to fill the gaps of 85.15% and 67.6%, respectively, of coal-fueled power plants in Shanghai and Korea. Otherwise, coal-fueled power plants will frequently act as buyers in the ETS market because the extremely low market price cannot stimulate coal-fueled power plants to further invest in pollution-reducing technology and improve environmental efficiency. In addition, although the difference in environmental efficiency between coal-fired power plants in Shanghai and in Korea is not large, the MAC difference reaches 73.7% due to the large difference in feed-in tariffs between these two regions. We suggest that policymakers should focus on the $CO_2$ MAC differences caused by huge feed-in tariff differences to avoid equity problems when building the structure of the Shanghai-Korea ETS cooperation. For instance, compared with power plants in Shanghai, policymakers should set a looser cap and a higher offset for Korean plants. To reduce the impact of feed-in tariffs on carbon trading in the market, it would also be effective to arrange a higher quota or lower carbon tax for coal-fueled power plants in Korea.

However, long-term trends cannot be identified in our study because we used cross-sectional data instead of panel data. Both Shanghai ETS and Korea ETS have been established for less than five years. When more panel data have been collected, future studies may be possible for more precise and valuable results.

**Author Contributions:** The authors are contributed each part of a paper by Conceptualization, Y.C.; Methodology,C.Q.; Software, C.Q.; Validation, Y.C.; Formal Analysis, C.Q.; Investigation, Y.C.; Resources & Data Curation, C.Q.; Writing-Original Draft Preparation, C.Q.; Writing-Review & Editing, Y.C.; Visualization, C.Q.; Supervision, Y.C.; Project Administration, Y.C.

**Funding:** This research received no external funding.

**Conflicts of Interest:** The authors declare no conflict of interest.

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
