# Peer review of "A Study of the Feasibility of International ETS Cooperation between Shanghai and Korea from Environmental Efficiency and CO2 Marginal Abatement Cost Perspectives"

_sustainability, doi:10.3390/su11164468_

Round 1

Reviewer 1 Report

This work examines an interesting problem related to a pilot international ETS program. The reviewer believes that the manuscript needs major revision.

First of all, the research method are not very detailed. For example, more explanation and justification of Equation 8 should be added. In subsection 4.1.1, the description of parameter estimation method was not provided. These missing details will make it very difficult for the readers to understand the study.

The research design needs significant improvement. The authors should show, in a concise and organized manner, the considerations on why the methods were chosen. Are there alternative methods? If there are other methods, can the authors use one as reference to compare the results?

Besides, there are many grammar errors or word choice problems in the paper. The authors need to thoroughly review their writing and correct the errors.

Author Response

Response on the Comments and Suggestions for Authors;

We really appreciate the kind, detailed comments for our paper, your comments made a great contribution to the improvement of our paper. Of course, we did our best to reflect all the comments by the reviewers as much as possible.

This work examines an interesting problem related to a pilot international ETS program. The reviewer believes that the manuscript needs major revision.

First of all, the research method are not very detailed. For example, more explanation and justification of Equation 8 should be added. In subsection 4.1.1, the description of parameter estimation method was not provided. These missing details will make it very difficult for the readers to understand the study.

Response: We fulfilled some intermediate missing link and explanations to the process of the calculations.

For example, we added some estimated equations.

Furthermore, we added an explanation to the parameter estimates as below:

Based on Equation (10), we solve the SFA problem by incorporating all the parameter estimate coefficients as shown in Table 2 into the Stata software program. We used the maximum likelihood method to estimate the stochastic frontier. All the variables obtained are consistent with the expectation and passed the statistical test. If the sign is negative, the negative effect of the variable in the production process is greater than the positive effect; on the contrary, if the sign is positive, the positive effect of the variable in the production process is greater than the negative effect.

The research design needs significant improvement. The authors should show, in a concise and organized manner, the considerations on why the methods were chosen. Are there alternative methods? If there are other methods, can the authors use one as reference to compare the results?

Response: We have refined the methodology section to make it more specific and easier for readers to understand. In addition, both DEA and LP have shortcomings that cannot be ignored, which have mentioned and explained in detail in our manuscript. The DEA method can only be applied when the function is differentiable everywhere. Moreover, there are different slopes of the efficient observation which located on the frontier, the different choice of slopes will lead to different results. As to LP approach, it ignores the random error and statistic noise.

Actually, in previous studies, some scholar did method comparative studies, and found that SFA was relatively precise and reliable.

Our methodology part mainly refers to the work of professor Rolf Fare in 2005, and we used the similar method did a previous research study (Yongrok, C., Chao, Q. Is the Korean ETS performance effective? Based on the MAC of coal-fueled power plants in Korea, Sustainability 2019, 11(9), 2504). In this paper, depending on the specific situation, we introduce Meta-frontier to solve group heterogeneity problems.

We mainly focused on the analysis of possible problems in the cooperation between Shanghai and Korea in the carbon trading market, so we did not conduct a comparative study of methods, and at the same time, we also want to avoid the lengthy problem of our paper.

Besides, there are many grammar errors or word choice problems in the paper. The authors need to thoroughly review their writing and correct the errors.

Response: Thank you for pointing out such an important problem. We have rechecked our manuscript several times and found out about 80 grammatical or word errors. All the mistakes we found have been corrected.

In addition, we have rechecked our paper in order to avoid other mistakes. Thanks for your scrupulous and professional comments again, it’s really helpful for the quality improvement of our paper.

Reviewer 2 Report

The purpose of this paper is to explore the feasibility of the pilot ETS cooperation between Shanghai and Korea from environmental efficiency and CO2 marginal abatement cost (MAC) perspectives. It estimates environmental efficiency and the CO2 MAC by using the directional distance function (DDF) and stochastic frontier analysis (SFA) for coal-fueled power plants. This is an interesting and important paper. In my opinion, this paper can be accepted for publication after the following revisions has been made:

(1) Suggest rewriting Abstract to briefly describe the motives, purposes, research methods, important results, limitations and future research directions of this research.

(2)  Suggest enhancing the descriptions of the reasons why chose Shanghai and Korea as the research objects.

(3) Suggest using a Table to describe the essentials and comparisons among the related researches mentioned in Literature Review.

(4) Suggest clearly defining the variables used in this paper, for example, CO2 in Equation (14).

(5) Suggest citing the related papers published in this journal “sustainability” to link this paper to the material of this journal “sustainability” (only one now).

(6) Suggest reformatting the References according to this journal’s format.

Author Response

Response on the Comments and Suggestions for Authors;

We really appreciate the kind, detailed comments for our paper, your comments made a great contribution to the improvement of our paper. Of course, we did our best to reflect all the comments by the reviewers as much as possible.

The purpose of this paper is to explore the feasibility of the pilot ETS cooperation between Shanghai and Korea from environmental efficiency and CO2 marginal abatement cost (MAC) perspectives. It estimates environmental efficiency and the CO2 MAC by using the directional distance function (DDF) and stochastic frontier analysis (SFA) for coal-fueled power plants. This is an interesting and important paper. In my opinion, this paper can be accepted for publication after the following revisions has been made:

(1) Suggest rewriting Abstract to briefly describe the motives, purposes, research methods, important results, limitations and future research directions of this research.

Response: We reorganized the abstract and added and removed some content.

Abstract: With the worldwide spread of ETS and the need for international cooperation on climate change, there is growing interest in linking ETSs. Along with the sustainable developmental strategy, the question of the environmental pollution, preventing and controlling are regarded as urgent affair by China and Korea. In the context of the great willingness of emission trading scheme (ETS) cooperation between Chinese government and Korean government, this paper examines the feasibility of the pilot ETS cooperation between Shanghai and Korea from environmental efficiency and CO2 marginal abatement cost (MAC) perspectives. We apply a directional distance function (DDF) and stochastic frontier analysis(SFA) to estimate environmental efficiency and the CO2 MAC for coal-fueled power plants in Shanghai and Korea with a cross-section data in 2015. The results indicate that the group frontier environmental efficiency of Shanghai and Korea reach a similarly high score. However, as to meta-frontier environmental efficiency, the coal-fueled power plants in Korea perform better than those in Shanghai. The CO2 MAC results indicate that with little gap in efficiency performance, the CO2 MAC of coal-fueled power plants is much higher than those in Shanghai due to the big feed-in tariff difference. Because the MAC not only relate to the environmental efficiency, but also relate to the feed-in tariff. Higher feed-in tariff leads to higher MAC. For this serious problem, refer to previous studies, we suggest that policymakers should focus on the huge CO2 MAC differences caused by huge feed-in tariff differences to avoid equity problems when build the structure of Shanghai-Korea ETS cooperation. For instance, compared with power plants in Shanghai, policymaker should set looser cap and more offset for Korean plants. To reduce the impact of feed-in tariff on carbon trading in the market, it is also an effective measure to arrange more quota or lower carbon tax for coal-fueled power plants in Korea. In addition, policymakers should fill gaps of 85.15% and 67.6% between realistic market price and the MAC results of coal-fueled power plants in Shanghai and Korea by introducing stricter regulation.

(2) Suggest enhancing the descriptions of the reasons why chose Shanghai and Korea as the research objects.

Response:In September 2016, China’s national development and reform commission, Korea’s Ministry of strategy and finance and Japan’s Ministry of the environment held a meeting in Beijing and decided to hold a regular meeting of carbon emission trading every year, through which they will discuss the establishment of a long-term unified market for carbon emission trading and unified emission testing methods, and seek a scheme for the exchange of carbon emission market among the three countries. As a more specific and substantive step, Korean ETS has signed a memorandum of understanding with China's Beijing environment exchange. Under the agreement, China and Korea will exchange market information and share experience on ETS operation. They agreed to expand cooperation on ETS, including joint efforts to link ETS in these two countries. In view of the great willingness of China and Korea government, the ETS cooperation between these two countries can be expected in the future.

The intense interest in the ETS cooperation between China and Korea, in addition to the geographical proximity, mainly due to their large number of international trade volume. Korea's carbon market is the first ETS among developing countries, while China is the world's largest carbon trading market. Both China ETS and Korea ETS got technological and institutional support from EU ETS which provides favorable conditions for compatibility between these two ETS markets.

(3) Suggest using a Table to describe the essentials and comparisons among the related researches mentioned in the Literature Review.

Response: We added a Table to compare the related researches.

Studies

Year

Method

Objective

Wang et al

2011

DDF/DEA

CO2 MAC of 28 provinces in China

Liu et al

2011

DF/DEA

CO2 MAC of 30 provinces in China

Du et al.

2012

DF/LP

CO2 MAC of 29 provinces in China

Zhang et al

2014

DDF/LP

CO2 MAC of 29 provinces in China

Wei et al

2013

DDF/SFA

CO2 MAC of the coal-fueled power plants in China

Oh et al

1999

DF/LP

Airborne pollutants MAC of Korea’s power plants

Lee

2010

DF/LP

CO2 MAC of Korea’s coal-fueled power plants

(4) Suggest clearly defining the variables used in this paper, for example, CO2 in Equation (14).

Response: We made some complementary explanations of the variables and coefficients in our paper.

Suggest citing the related papers published in this journal “sustainability” to link this paper to the material of this journal “sustainability” (only one now).

Response: We have three references from “sustainability” now.

Zhang, N., Choi, Y. A note on the evolution of directional distance function and its development in energy and environmental studies 1997–2013. Renew. Sustain. Energy Rev 2014. 33, 50–59.

Yongrok, C.; Chao, Q. Is the Korean ETS performance effective? Based on the MAC of coal-fueled power plants in Korea, Sustainability. 2019, 11(9), 2504

Lee, H.; Choi, Y. Environmental Performance Evaluation of the Korean Manufacturing Industry Based on Sequential DEA. Sustainability. 2019, 11, 874.

Suggest reformatting the References according to this journal’s format.

Response: We modified the formation mistake in the reference part.

In addition, we have rechecked our paper in order to avoid other mistakes. Thanks for your scrupulous and professional comments again, it’s really helpful for the quality improvement of our paper.

Round 2

Reviewer 1 Report

The authors have addressed the issues pointed out by the reviewer. The paper, after some minor corrections of language and formatting, is acceptable to be published in Sustainability.